# Low-Complexity Hyperbolic Embedding Schemes for Temporal Complex Networks

**DOI:** 10.3390/s22239306

**Published:** 2022-11-29

**Authors:** Hao Jiang, Lixia Li, Yuanyuan Zeng, Jiajun Fan, Lijuan Shen

**Affiliations:** 1School of Electronic Information, Wuhan University, Wuhan 430072, China; 2Wuhan Digital Engineering Institute, Wuhan 430074, China

**Keywords:** dynamic network embedding, hyperbolic space, matrix perturbation, maximum likelihood estimation

## Abstract

Hyperbolic embedding can effectively preserve the property of complex networks. Though some state-of-the-art hyperbolic node embedding approaches are proposed, most of them are still not well suited for the dynamic evolution process of temporal complex networks. The complexities of the adaptability and embedding update to the scale of complex networks with moderate variation are still challenging problems. To tackle the challenges, we propose hyperbolic embedding schemes for the temporal complex network within two dynamic evolution processes. First, we propose a low-complexity hyperbolic embedding scheme by using matrix perturbation, which is well-suitable for medium-scale complex networks with evolving temporal characteristics. Next, we construct the geometric initialization by merging nodes within the hyperbolic circular domain. To realize fast initialization for a large-scale network, an R tree is used to search the nodes to narrow down the search range. Our evaluations are implemented for both synthetic networks and realistic networks within different downstream applications. The results show that our hyperbolic embedding schemes have low complexity and are adaptable to networks with different scales for different downstream tasks.

## 1. Introduction

Most real-world networks are complex networks with small-world, scale-free and strong-clustering properties. Complex network embedding is a valid tool for downstream network analysis tasks [1,2,3]. Many network embedding approaches based on Euclidean space have been addressed well  [4,5,6,7,8,9,10,11,12,13,14,15,16,17,18,19]. Complex networks usually have latent tree-like and scale-free properties [20]; Euclidean space mapping can not capture the above features well. According to this, some researchers propose non-Euclidean network embedding [21,22]. They have shown that the hyperbolic space is more suitable for complex network representation with a tree-like hierarchical organization [20]. The hyperbolic space extracts the hierarchical topology organization by approximating tree-like structures smoothly with constant negative curvature rather than the flat Euclidean space [23]. The network hyperbolic embedding theory makes the geometrical representation of the complex network while it preserves the small-world and scale-free properties well. It can effectively interpret the hierarchical topology characteristics and generation mechanisms of complex networks. Compared with network embedding in Euclidean space, dynamic hyperbolic space embedding is a research area not yet fully studied. Some of the hyperbolic space embedding approaches are rather highly complex.

Current hyperbolic embedding approaches are mainly divided into three categories. The first category is based on manifold learning. The research work [24] proposes a data-driven manifold learning approach based on Laplacian network embedding. The approach is similar to Laplacian matrix decomposition in Euclidean space by using the Laplacian matrix for eigenvalue decomposition. The second category is based on the maximum likelihood estimation approach. HyperMap [25] infers angular coordinates by reproducing the generation of a network generation model. All nodes are sorted in descending order. The possible angle values are traversed to maximize the likelihood function for the most suitable angular coordinates. HyperMap-CN [26] proposes to derive the hidden geometric coordinates of nodes in a complex network based on the number of common neighborhoods. It utilizes the common neighbor information from the likelihood function of the HyperMap method to improve the accuracy of node embedding coordinates. The third category is the hybrid approach that combines both manifold learning and maximum likelihood estimation. Although LaBNE has high embedding efficiency, it sacrifices embedding performance. Similarly, HyperMap has high embedding performance, but the complexity is also high. Accordingly, LaBNE+HM [27] proposes using LaBNE for network embedding to obtain initial embedding values of nodes, and it utilizes HyperMap to obtain the final embedding coordinates by sampling the angles near initial values.

Nevertheless, most of the aforementioned methods are designed for static networks. In the real world, networks have inherent dynamics with evolving characteristics. For example, nodes in social networks add and delete neighbors with varied social relations. The nodes in brain networks make changes to the neighboring relations according to new connections of neurons. However, efficient representations with varied nodes and edges are extremely crucial, especially for the stabilization evolution of network application scenarios [28]; it presents challenges for the embedding of dynamic networks.

Inspired by Node2vec [10], which extended Deepwalk by changing the random walk method, researchers introduced temporal meta-paths [29,30] to modify the sampling method. Both approaches are derivatives of static approaches, which do not capture the dynamics and high-order proximity of nodes and edges of a local structure well. The high-order proximity has proven to be valuable in capturing the network structure [31]. Research in [32] proposes to separate the dynamic network into several snapshots and then process the static network embedding according to the variation. Inevitably, some complex terms involving global structural information occur in the process of preserving global higher-order approximations, which results in high complexity. With this dilemma, some researchers propose conducting dynamic embedding with the consideration of network evolution [32,33,34,35]. They propose to capture characteristics with variations to reflect network dynamics and then improve the efficiency of application tasks based on features. Cao [33] et al. made a review on current dynamic network embedding approaches. They point out that current embedding approaches have made breakthroughs in many ways, but there are still problems to be solved. For example, how to effectively capture the influence of node variations on neighboring nodes and the local network structure are still key problems. How to overcome problems such as data storage, training efficiency and heterogeneous information fusion [36] for large-scale network embedding is also still not well addressed.

According to the above, the main challenges for hyperbolic space embedding of temporal complex networks include: (1) The embedding complexity of hyperbolic space is a key factor for complex network analysis efficiency. (2) Dynamic network embedding needs to be adaptive toward variations within network evolution.

In this paper, we propose low-complexity hyperbolic embedding schemes for temporal complex networks. First, we propose a low-complexity hyperbolic embedding approach using matrix perturbation with time-evolving features for a medium-scale complex network. Next, we propose a fast update hyperbolic embedding approach with a local maximum likelihood estimation-based geometric initialization and R tree-based local search for large-scale complex networks.

The main contributions of this paper are summarized as follows:(1)We propose MpDHE to implement dynamic network hyperbolic embedding with low complexity, i.e., $O(T\left(n+d_{x}+l_{x}\right)+n^{2})$. To the best of our knowledge, we are the first to model the increment of the network utilizing the matrix perturbation when inferring hyperbolic coordinates.(2)We computed geometric initialization to embed medium-scale dynamic networks via hyperbolic circular domain construction.(3)We implement the proposed schemes for real-world network scenarios with several kinds of downstream application tasks, including community discovery, visualization and routing, which proves the efficiency and effectiveness.

The remainder of the paper is organized as follows. Section 2 gives some preliminaries for hyperbolic embedding. Section 3 proposes a novel low-complexity embedding scheme for dynamic temporal complex networks. Section 4 gives the performance evaluations. Section 5 concludes the paper.

## 2. Some Preliminaries

### 2.1. Complex Network and Generation Model

In a realistic world, many complex systems can be represented by networks with a collection of nodes and edges, i.e., G=(V,E). Different from small-scale networks, most complex systems are large-scale networks following a power-law distribution. They are modeled as temporal complex networks within the time evolution processes. Time is divided into continuous time steps, which then form the sequence of network snapshots for each time step. The temporal complex network can be represented by: $G^{\left(t\right)}=\left(V^{\left(t\right)},E^{\left(t\right)}\right)$. For each time step *t*, the adjacency matrix is denoted as $A^{\left(t\right)}$. The element in $A^{\left(t\right)}$ is 1 if there is an edge between node *i* and *j*; otherwise it is 0. The Laplacian matrix of the graph is: $L^{\left(t\right)}=D^{\left(t\right)}-A^{\left(t\right)}$, where *D* is a matrix with node degrees on its diagonal (with 0 elsewhere). We assume all the networks considered in the paper are connected networks. For unconnected networks, each of the connected subnetworks is taken into consideration. In this case, *A*, *D* and *L* are symmetric matrices, and the degree of each node is positive.

Two commonly used complex network generation models are the popularity-similar optimization (PSO) model [37] and the nonuniform PSO (nPSO) model [38]. The PSO model keeps a trade-off between node generation time and node similarity. The node generation time is positively related to node popularity. The PSO model can generate a complex network of *N* nodes with real, known hyperbolic coordinates. The model parameters include an average node degree 2m, the scaling exponent γ and the network temperature *T*. The PSO model can simulate how random geometric graphs grow in the hyperbolic space, generating realistic networks with small-world, scale-free and strong-clustering features. PSO cannot reproduce the community structure of a network, and the nPSO model is proposed for community structure based on this. It enables heterogeneous angular node attractiveness by sampling the angular coordinates from a tailored nonuniform probability distribution, e.g., a mixture of Gaussians. The nPSO model can explicitly determine the number and size of community structures with an adjustment of network temperature, which controls network clustering and generates highly clustered networks efficiently.

### 2.2. Hyperbolic Space and Poincare Disk Model

Hyperbolic space is hard to imagine and equivalently embedded into Euclidean space. Hyperbolic space is even “larger” than Euclidean space. In this paper, we use the Poincare disk model as the embedding target. The circumference and area equation of a hyperbolic disk with hyperbolic radius *R* and centroid as (0,0) can be represented by (Equation 1) and (Equation 2).
(1)L=2πsinh(R)
(2)A=2π(cosh(R)−1)
where both the circumference and the area present an exponential increase along radius *R* (i.e., $\sinh x=\frac{e^{x}-e^{-x}}{2}$, $\cosh x=\frac{e^{x}+e^{-x}}{2}$). The hyperbolic space grows rapidly along the radius. The region grows to a fairly large space at the edge of the disk, which is the most prominent feature of the hyperbolic space.

Hyperbolic space is suitable to represent complex networks, especially for tree-like property-based structures. Actually, hyperbolic space can be viewed as the continuous version of a tree-based network. For a *n*-numeration tree in the network system, the circumference and the area of the Poincare disk correspond to the number of nodes within *s*-hop from the root as: $\left(n+1\right)n^{s-1}$ and the total number of nodes within *s*-hop from the root as: $\frac{\left(n+1\right)n^{s}-2}{n-1}$, respectively. If the curvature of the hyperbolic space satisfies $\zeta =\sqrt{\left|K\right|}=\ln n$, then the circumference and the area of the hyperbolic space increase with the rate of $e^{\zeta r}$, consistent with the growth rate $n_{r}$ of the *n*-numeration tree. In this case, the tree structure can be regarded as a discrete hyperbolic space, which is shown in Figure 1.

Branches of the tree structure need a storage space of exponential magnitude, which is well supported by hyperbolic geometry methodology. The scale-free and tree-like structure of complex networks fit with the negative curvature and the exponential expansion of hyperbolic space, so the hyperbolic space embedding approaches are well suited for geometry-based representation learning of complex networks. For hyperbolic space embedding, the radial coordinates in the Poincare disk represent the popularity of nodes. The angular coordinates represent the node similarities. Moreover, we can effectively illustrate the evaluation of complex networks as the completion between popularity and similarity by using the Poincare disk model. Further, the Poincare disk model is also effective for explaining the topology features of complex networks.

### 2.3. Initial Static Embedding

To mine the evolution characteristics of the network and reduce the complexity, we utilize the Laplacian matrix decomposition-based hyperbolic embedding approach LaBNE in  [24] to make an initial static embedding for the network snapshot at time step $t_{0}$. We then update the network embedding results for the subsequent time steps by capturing the main variations in the topology structure. The complexity of hyperbolic embedding mainly comes from angular coordinate embedding, so the embedding for temporal complex networks focuses on the update of angular coordinates.

**LaBNE for initial static embedding**: The common sense of the hyperbolic network model is that the connection probability between nodes is negatively correlated with the angle difference. The connected nodes have similar angles. In LaBNE, the network is embedded into the two-dimensional hyperbolic plane $H^{2}$ represented by a Euclidean circle, which gives matrix *Y* with shape N×2 as: $Y=[y_{1},y_{2}]$. In which each row represents the embedding coordinate of the node. By using the Laplace operator, the objective is to minimize $tr\left(Y^{T}LY\right)=\frac{1}{2}\sum_{i,j}a_{ij}\left|\right|Y_{i}-Y_{j}{\left|\right|}^{2}$. Where trace is the weighted sum of the distance between adjacent nodes. By minimizing the trace, it can reduce the Euclidean distance between connected nodes. If nodes are distributed around a circle centered at the origin point, then the distance in Euclidean space also reflects the angle difference. To avoid being arbitrarily scaled, the problem also includes an additional constraint as: $Y^{T}Y=I$. The optimization problem can be described as:(3)$$\min tr(Y^{T}LY)s.t.Y^{T}Y=I$$

Using the Rayleigh–Ritz theorem, the solution of this problem consists of eigenvectors corresponding to two minimum non-zero eigenvalues of the generalized eigenvalue problem: $L^{\left(t\right)}Y=\lambda D^{\left(t\right)}Y$. Where the minimum eigenvalue is zero, we take eigenvectors corresponding to the second- and third-smallest eigenvalues.

Embedding the network in a two-dimensional hyperbolic disk needs radial coordinates and angular coordinates of nodes. Angular coordinates can be approximated by: $\theta =\arctan\left(\frac{y_{2}}{y_{1}}\right)$ according to the conformal properties, where $y_{1}$ and $y_{2}$ are the first and second items of the row vectors corresponding to the nodes in *Y*. In addition, we need an additional equidistant adjustment step to distribute nodes on the disk evenly. There are two ways to calculate the radial coordinate by utilizing the PSO model and static estimation [39]. We choose the latter to get the radius of hyperbolic disk *R* and the radial coordinate r(i), which can be calculated by (Equation 4) and (Equation 5).
(4)$$R=2\ln\left(\frac{4n^{2}\alpha^{2}T}{\left|E\right|\cdot \sin\left(\pi T\right){\left(2\alpha -1\right)}^{2}}\right)$$
(5)$$r\left(i\right)=\min\left\{R,2\ln\left(\frac{2n\alpha T}{deg\left(i\right)\cdot \sin\left(\pi T\right)\left(\alpha -\frac{1}{2}\right)}\right)\right\}$$
where *n* is the total number of nodes in the network (the maximum connected subgraph), $\alpha =\frac{\gamma -1}{2}$ and γ is the power-law distribution exponent. *T* is the clustering coefficient of the network. |E| is the number of edges.

The calculation radial coordinate embedding has complexity O(n). The calculation of angular coordinate embedding needs the first two items of the generalized eigen decomposition with complexity $O(n^{2})$.

## 3. Hyperbolic Embedding Schemes for Temporal Complex Networks

We propose dynamic hyperbolic embedding schemes to tackle the challenges of complexity and dynamics in the temporal complex network. First, we propose the matrix perturbation-based dynamic network hyperbolic embedding scheme (MpDHE) to achieve low complexity, which is adaptive within the scale-fixing network. We then generalize the implementation of MpDHE into a large-scale network utilizing the geometry initialization in a hyperbolic circular domain. Figure 2 shows the overview of the proposed scheme for dynamic hyperbolic embedding.

### 3.1. MpDHE Scheme

To reduce the time complexity for dynamic hyperbolic embedding, we propose the use of matrix perturbation [32,40] to update the embedding coordinates in the MpDHE scheme. Matrix perturbation is also commonly used in the dynamic network embedding of Euclidean space. Compared with the matrix at time step *t*, a matrix perturbation is involved in the matrix at time step t+1. For a specific feature dimension *i* with feature pair as: $(\lambda_{i},y_{i}),i=1,2,\ldots ,d$, the generalized eigenvalue problem after the perturbation is shown in (Equation 6).
(6)$$\left(L^{\left(t\right)}+\Delta L\right)\left(y_{i}+\Delta y_{i}\right)=\left(\lambda_{i}+\Delta \lambda_{i}\right)\left(D^{\left(t\right)}+\Delta D\right)\left(y_{i}+\Delta y_{i}\right)$$

According to the matrix perturbation, the approximate solutions of the increments of eigenvalues and eigenvectors are shown in (Equation 7) and (Equation 8).
(7)$$\Delta \lambda_{i}=y_{i}^{\prime }\Delta Ly_{i}-\lambda_{i}y_{i}^{\prime }\Delta Dy_{i}$$
(8)$$\Delta y_{i}=-\frac{1}{2}y_{i}^{\prime }\Delta Dy_{i}y_{i}+\sum_{j=2,j\neq i}^{3}\left(\frac{y_{j}^{\prime }\Delta Ly_{i}-\lambda_{i}y_{j}^{\prime }\Delta Dy_{i}}{\lambda_{i}-\lambda_{j}}\right)y_{j}$$

The objective function of network embedding in Euclidean space is the same as the objective of solving angular coordinates in hyperbolic embedding; the above scheme can be directly used to incrementally update angular coordinates in dynamic hyperbolic embedding. Specifically, the embedding vectors in Euclidean space are deduced as follows:(9)$$\begin{array}{ccc} y_{i}^{\left(t+1\right)} & = & y_{i}^{\left(t\right)}+\Delta y_{i} \end{array}$$(10)$$\begin{array}{ccc} \lambda_{i}^{\left(t+1\right)} & = & \lambda_{i}^{\left(t\right)}+\Delta \lambda_{i} \end{array}$$

Then, according to the conformal property, the angular coordinates $\Theta^{\left(t\right)}={\left\{\theta_{j}^{\left(t\right)}\right\}}_{1\times n}$ at time *t* can be calculated by the corresponding embedding vectors ${\left\{y_{i}^{\left(t\right)}\right\}}_{i=2,3}$ as follows:(11)$$\Theta^{\left(t\right)}=\arctan\frac{y_{2}^{\left(t\right)}}{y_{3}^{\left(t\right)}}$$

Obviously, the differences in the network between continuous time steps induce changes in the embedding vectors, which are analytically formulated with the incremental eigen decomposition. Therefore, the dynamic hyperbolic embedding at a later time step can be implemented in low complexity. Specifically, the time complexity of MpDHE is analyzed as follows, and the framework is summarized as Algorithm 1.

**Time complexity analysis**: Suppose *T* is the total number of time steps to be predicted. Each time the radial coordinate embedding complexity is the same as LaBNE, i.e., O(n), then the total complexity of the radial coordinate calculation is O(nT). For angular coordinate embedding, the complexity of the initial value setting is $O(n^{2})$. The complexity of updating eigenvalues is $O(d_{x}+l_{x})$, and the complexity of updating *k* eigenvectors is $O(n+d_{x}+l_{x})$. Where $d_{x}$ and $l_{x}$ are the numbers of non-zero items in sparse matrices ΔD and ΔL, respectively. In general, the complexity of MpDHE is $O(T\left(n+d_{x}+l_{x}\right)+n^{2})$. For $d_{x}\ll n$, $l_{x}\ll n$, the MpDHE scheme effectively reduces the embedding complexity.
**Algorithm 1: **MpDHE algorithm
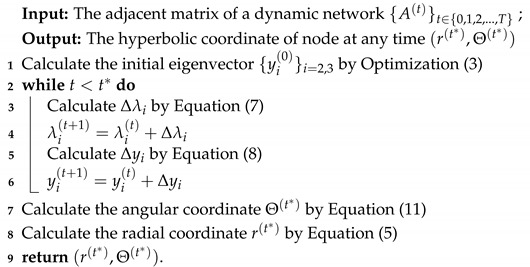


### 3.2. Geometric Initialization

However, the matrix perturbation in the above MpDHE cannot embed new nodes in the network. The basis of matrix perturbation makes updates based on the eigenvectors from the previous time step. For the node not existing at the previous time step *t* but appearing at time step t+1, MpDHE is not suitable for this case. It cannot update for no eigenvectors from the previous time step.

For this reason, we technically use eigenvectors at time step *t* to calculate values for new nodes appearing at t+1 and construct a geometric initialization by hyperbolic circular domain.

Obviously, the hyperbolic distance between nodes determines the connection probability of them in the hyperbolic disk. The shorter the hyperbolic distance between the two nodes, the bigger the connection probability is, as well as the similarity. When a new node occurs, if the original node is far from the new one, it has less effect on the angular coordinate. The initial angular coordinates can, therefore, be calculated from those nodes with a small hyperbolic distance from the new node.

Based on the above, we first filter out the neighbors with small and similar degrees toward the new node and set the mean of their angular coordinates as a basic approximation of the new node. The corresponding computational process is shown as (Equation 12).
(12)$${\hat{\theta }}_{i}^{\left(t+1\right)}=\frac{\sum_{j\in \Psi (i)}\theta_{j}^{\left(t\right)}}{m}$$
where Ψ(i) is the neighboring node set toward the new node. *i* is a small and similar degree, and *m* is the number of nodes in the set.

Then we calculate the hyperbolic circular domain with the basic approximation as the center and select nodes inside the circle. The distance $d_{ij}$ between node $\left(r_{i},\theta_{i}\right)$ and node $\left(r_{j},\theta_{j}\right)$ is shown in (Equation 13) by using the hyperbolic cosine theorem.
(13)$$\cosh\left(d_{ij}\right)=\cosh\left(r_{i}\right)\cdot \cosh\left(r_{j}\right)-\sinh\left(r_{i}\right)\cdot \sinh\left(r_{j}\right)\cdot \cos\left(\Delta \theta_{ij}\right)$$

Given the centroid of the hyperbolic disk with radius *R* as: $\left(r_{0},\theta_{0}\right)$, the radial coordinate $r_{h}$ corresponding to the angle $\theta_{h}$ on the disk is shown in (Equation 14). Where $c=\frac{e^{R}+1}{e^{R}-1}$.
(14)$$\begin{array}{c} {\left(r_{h}\cos\left(\theta_{h}\right)-\frac{r_{0h}\cos\left(\theta_{0h}\right)\left(c^{2}-1\right)}{c^{2}-r_{0h}^{2}}\right)}^{2}+(r_{h}\sin\left(\theta_{h}\right)-\\ \frac{r_{0h}\sin\left(\theta_{0h}\right)\left(c^{2}-1\right)}{c^{2}-r_{0h}^{2}}{)}^{2}={\left(\frac{(r_{0h}^{2}-1)c}{r_{0h}^{2}-c^{2}}\right)}^{2} \end{array}$$

The hyperbolic circle is not easy to represent in an equation directly. To quickly find the inner range of the hyperbolic disk by using R-tree (Rectangle tree) [41], we sample points on the hyperbolic disk and then outline the polygon contour to approximate the hyperbolic disk. R-tree is a tree-based data structure that can be used for spatial high-dimensional data storage and fast query. The core strategy of R-tree is to aggregate adjacent nodes and use the minimum circumscribed rectangles of them as the nodes of each layer in the tree. R-tree can query node sets inside the polygon quickly.

Here we take the polygon contour obtained by outlining the hyperbolic disk as the input of the R-tree. We then approximately query the node sets of the hyperbolic disk. Considering the error that exists from transforming the tanh(x) function and R-tree search based on the maximal circumscribed rectangle, more nodes will certainly be searched than the exact result. However, we can still think that nodes beyond the searched results have a distance of more than *R* from the center of the disk. Therefore, the nodes beyond the search results can be ignored, i.e., the search results of R-tree are within the initialization range. The geometric initialization is calculated as follows:(15)$$\theta_{i}^{\left(t+1\right)}=\frac{\sum_{j\in GI({\hat{\theta }}_{i}^{\left(t+1\right)})}\theta_{j}^{\left(t\right)}}{\left|GI({\hat{\theta }}_{i}^{\left(t+1\right)})\right|}$$
where $GI({\hat{\theta_{i}}}^{\left(t+1\right)})$ is the node set contained within the hyperbolic circular domain. $\left|GI({\hat{\theta }}_{i}^{\left(t+1\right)})\right|$ is the number of nodes in the set.

## 4. Performance Evaluations

In this section, we verify the performance of our schemes with numerous evaluations. First, we perform evaluations on the reliability of our schemes by comparison with the eigen decomposition-based scheme in terms of MSE. Then, we perform evaluations on synthetic networks by comparisons among the other static hyperbolic embedding schemes in terms of embedding precision. Afterward, we implement our schemes with different downstream tasks by comparison with the other hyperbolic embedding schemes.

### 4.1. Scheme Analysis

The proposed MpDHE is combined with the matrix perturbation and conformal mapping to reduce the dynamic embedding complexity with preserving the embedding precision. Conformal mapping transforms a Euclidean coordinate into a hyperbolic coordinate, which is lossless. However, if matrix perturbation acts on quickly re-embedding, it would inevitably incur errors.

To analyze the reliability of the Euclidean coordinates obtained by matrix perturbation, we implement ablation experiments on 10 groups of synthetic networks. These groups of networks are constructed by nPSO with network scales ranging from 100 to 1000. The proportion of changed nodes between net0 and net1 is 5%. We calculate the mean square error (MSE) between eigenvectors obtained from eigen decomposition and matrix perturbation. The corresponding results are shown in Figure 3.

It shows that all the MSEs are at a low level (under 0.05). Moreover, the MSE decreases with the increase in nodes. It indicates that the proposed method is capable of embedding large-scale networks under a convergence error.

### 4.2. Embedding Performance Evaluations

#### 4.2.1. Settings

To verify the efficiency effectiveness of our embedding schemes for complex networks with different parameters, we generate 10 PSO synthetic networks with the combination of parameters as: T=0.1,0.4,0.7,1, 2m=4,6,8,10 and γ=2,2.5,3. The network scale is set as 10,000. We then implement LaBNE and the two proposed embedding schemes to embed the network into the Poincare disk. The results are based on the average of ten networks with the above parameter configurations.

To perform evaluations within network dynamic scenarios, we generate two snapshots of networks with the above generation configurations. It can also be easily extended to more snapshots. The first snapshot represents the initial “net0”, and the second snapshot represents “net1”. The second snapshot has a 1% change in nodes compared to the first snapshot, i.e., newly added nodes have the ratio $\frac{1}{m+1}$, and the varied old nodes have the ratio $\frac{m}{m+1}$ according to the PSO model.

We perform evaluations on our MpDHE scheme with the performance bounds provided by the other static embedding schemes for dynamic network situations. Those static embedding schemes can still be used for dynamic scenarios if we treat each snapshot of the dynamic network as the static embedding scenario. Obviously, this procedure would incur high complexity, and it is not realistic for large-scale network application scenarios. In our evaluations, improved static embedding schemes are used as a precision bound for comparisons.

#### 4.2.2. Baselines

In performance evaluation experiments, we compare the performance of the following static embedding methods to evaluate their effectiveness.

**EE [42]:** EE is an efficient hyperbolic embedding method with a greedy strategy, which combines common neighbor information with the maximum likelihood of optimizing the embedding.

**Coalescent [43]:** Coalescent approximates the hyperbolic distance between connected nodes with two manifold-learning-based pre-weighting strategies. The final embedding vectors are adjusted via maximum likelihood.

**LPCS [44]:** LPCS is a novel hyperbolic embedding method utilizing the community information of the network; it embeds nodes from a common community to preserve the mesoscale structure of the networks.

**CHM [45]:** CHM detects communities of the network and then constructs a fast index to solve the maximum likelihood with the guide of the obtained communities.

**Mercator [46]:**
Mercator embeds networks into the $S^{1}$ model incorporating machine learning and maximum likelihood via a fast and precise mode.

**LaBNE [24]:** LaBNE is a manifold learning method based on the Laplace eigen decomposition. Embedding vectors are transformed into a two-dimensional hyperbolic plane according to conformal mapping.

#### 4.2.3. Metrics

We use two metrics to evaluate the network embedding: the hyperbolic distance correlation and the concordance score.

*Hyperbolic distance correlation (HD-corr):* HD-corr is the Pearson correlation of the pairwise hyperbolic distance between initial coordinates and embedding coordinates. The Pearson correlation coefficient can measure the linear relation between two objects and estimate whether the linear relation can be fitted to a straight line. The closer the absolute value approaches 1, the stronger the correlation is. Otherwise, the closer the absolute value approaches 0, the weaker the correlation is.

*Concordance score (C-score):* C-score is the proportion of node pairs arranged in the same rotation direction of the initial network versus the embedding network, which is shown in (Equation 16).
(16)$$C-score=\frac{\sum_{i=1}^{n-1}\sum_{j=i+1}^{n}\delta \left(i,j\right)}{n\cdot \frac{\left(n-1\right)}{2}}$$

In which *n* is the node number and *i* and *j* represent two nodes. If the direction (clockwise or anti-clockwise) of the shortest angular distance between *i* and *j* in the initial coordinates is the same as that in the embedding coordinates, then δ(i,j) is 1. Otherwise, δ(i,j) is 0. Similar to HD-corr, the C-score increases from 0 to 1 as the embedding performance improves. Therefore, these two metrics can guide the parameter selection of MpDHE. Specifically, we set *T* and γ to high HD-corr and C-score.

#### 4.2.4. Results

First, we evaluate the embedding performance within different complex network configuration parameters. Figure 4 shows the mean HD-corr within different complex network parameters, where each row represents one specific embedding scheme. Each column corresponds to a specific power-law index. For each subgraph, the horizontal axis represents the temperature coefficient *T*, and the vertical axis represents the average degree *m*. The color of the heat map reflects the HD-corr value. The embedding performance increases as the color deepens. The results show that the embedding performance of our scheme is pretty good when compared with the bound of LaBNE within different network configurations. In addition, the optimal network configuration of parameter combinations appears at the lower left corner of the heat map for schemes, i.e., the parameter combination of T=0.1, 2m=10 and γ=3. This indicates that the hyperbolic embedding scheme is well-suited for low-temperature and dense, complex networks.

To further verify the adaptability of our embedding scheme within different networks, we perform evaluations on three groups of experiments in different scenarios.

*The first group*: We set the complex network configuration as a parameter combination of: T=0.1, 2m=10 and γ=3. The initial network size is set to 10,000 nodes. Then, we choose a different proportion of changed nodes in the network, and generate 10 groups of networks. The embedding results of HD-corr and C-score are shown in Table 1. Compared with the other static embedding methods, our scheme has pretty good embedding performance. As the proportion of changed nodes increases, the HD-corr and C-score of the embedding schemes slightly decrease. At the same time, C-score is higher than HD-corr within the same situation since HD-corr measures hyperbolic distance, while C-score only measures the relative angle.

*The second group*: We set the complex network configuration parameters as the same as the first group; we only changed the node variation ratio to 1%. Table 2 shows the HD-corr and C-score with a network scale from 1000 to 20,000. The results also show that our scheme achieves good embedding performance when compared with other methods.

*The third group*: We set the network scale as 10,000 nodes and the node variation ratio as 1%. Then, we extended the network time step from two steps to six steps, i.e., net0, net1 till net5. The other network configuration parameters are the same as the previous two groups. The results are shown in Table 3. We find that the HD-corr and C-score of our method slightly decreases but still stay above 0.96 and 0.99 with more updates.

From the three groups of evaluations, we find that our dynamic embedding scheme is rather competitive and may even be superior when compared with many schemes of static embedding in terms of embedding efficiency. Coalescent is the only one that obviously exceeds our scheme. It shows that our dynamic updating process does not incur an obvious loss of embedding efficiency.

### 4.3. Embedding Efficiency Evaluations

#### 4.3.1. Settings

We fix the proportion of changed nodes to 1%, change the network scale from 2500 to 17,500 in the initial network net0 and use LaBNE and MpDHE to make the embeddings for net1. The other parameters are set as: T=0.1, 2m=10 and γ=3.

We make embedding efficiency evaluations on our proposed hyperbolic embedding scheme with both a synthetic network generated by the nPSO model and realistic, complex network datasets. To embody the dynamic network scenario, we involve two continuous time steps of network scenarios in the evaluation, which can also be easily extended to more continuous time steps. The dynamic network situation includes two continuous time steps: the initial “net0” and the second time step of “net1”. We compare the embedding time of different hyperbolic embedding schemes to “net1”. The details of the complex networks used for the evaluation are as follows.

nPSO-1: a synthesis network dataset generated by the nPSO model. We set 15 communities, and the network parameters are n=1000,m=5,T=0.1,γ=3. The proportion of varied nodes between net0 and net1 is 20%.

Students: Dataset [47] includes the relationship between some students at a French high school for a 5-day duration. The dataset is available at http://www.sociopatterns.org/datasets/high-school-contact-and-friendship-networks/, accessed on 7 September 2022.

BS: Network [48] consists of recorded users’ behavior on the Internet in a Chinese city for two weeks. It uses base stations (BS) as nodes. If there are people exchanges over a period of time, we think the two base stations have an edge.

DBLP: Is the reference network of DBLP [49], which is a database of scientific publications. The dataset can be downloaded from http://konect.cc/networks/dblp-cite/, accessed on 7 September 2022.

arXiv-HepPh: Is a co-reference network of scientific papers from the high-energy physics and phenomenology (Hep-Ph) part of arXiv [50]. The dataset can be downloaded from http://konect.cc/networks/ca-cit-HepPh/, accessed on 7 September 2022.

The statistical metrics of the complex network generated by the nPSO model and from the realistic dataset are shown in Table 4. Where $\left|V(G_{0})\right|$ is the number of nodes in the initial network net0. $\left|E(G_{0})\right|$ is the number of edges in the initial network net0. $\left|\gamma (G_{0})\right|$ is the power-law distribution index in the initial network net0. $|E_{add}|$ is the number of newly added edges in net1 compared with net0. Correspondingly, $|E_{del}|$ is the number of deleted edges in net1 compared with net0.

#### 4.3.2. Results

The embedding time results are shown in Table 5. We can see that our scheme achieves the best time efficiency within the different datasets. The static hyperbolic embedding schemes need to recompute coordinates each time the network changes, and our scheme only needs to update coordinates with low time complexity. In the last section, Coalescent achieves the best result; it exceeds our scheme a little bit. However, our scheme takes less time and achieves very close results, so our scheme is effective with better embedding time efficiency and valid precision.

### 4.4. Visualization Effect for Downstream Community Discovery

#### 4.4.1. Settings

We perform performance evaluations of our proposed hyperbolic embedding schemes within the downstream community discovery task. Our evaluations are implemented on both complex networks generated by the nPSO model and the realistic complex network datasets.

#### 4.4.2. Results

We implement our proposed scheme and then utilize the Critical Gap Method (CGM) [51] to perform the downstream community discovery task based on the embedding. We show the visualization of our scheme when compared with the classic community discovery algorithm: Louvain algorithm [52]. It forms the community directly based on the topology structure without representation of learning-based approaches. The performance of the community discovery is quantified by modularity in Table 6. Evidently, the modularity obtained by MpDHE still maintains a high level on most networks.

To conveniently show the visual effect of networks based on our schemes, we choose two medium-scale networks. One is from the synthetic network, and the other is from the real network.

*Visualization of the synthetic network:*Figure 5 and Figure 6 show the visualization effect of the community discovery task for net0 on the nPSO-1 dataset. Figure 5 is the visualization effect of the community discovery task using the Louvain algorithm. In the figure, the color represents community division, and the same color represents the same community. Figure 6a shows the hyperbolic embedding coordinates of the network, and Figure 6b is the result of community discovery using the CGM algorithm based on the embedding coordinates in Figure 6a.

Comparing the above figures, we can see that the Louvain algorithm forms the community based on network topology, so the location of nodes and the distance between nodes have no obvious physical meaning. However, community discovery based on the hyperbolic embedding results in a good visualization such that the embedding node location represents the balance between popularity and similarity. The similar color of two communities (i.e., with similar angular coordinates) in the figure implies the similarity of the two communities. Moreover, the radial coordinates (representing the popularity) imply the key nodes (circled in Figure 6b) in the community.

*Visualization for the realistic complex network:* For the realistic, complex network, we choose the “Students” dataset to show the visualization effect. Figure 7 and Figure 8 show the visualization results of the community discovery of net0 in the “Students” dataset. Figure 7 shows the visualization of community discovery using the Louvain algorithm. The same color in the figure represents the same community. Figure 8c shows the hyperbolic embedding coordinates of the network. Similar to the nPSO network, the visualization effect based on our embedding scheme is obviously better than the Louvain algorithm. The interaction within and between communities is obvious and clear. If the labels and attributes of nodes are known, such visualization effects will be further improved.

We then exhibit the visualization effect of the “Students” dataset within two continuous time-step scenarios. Figure 8a,b shows the evolution of net0 and net1 in the “Students” dataset. The results show that the community interactions vary with the two time steps’ evolution, but the overall community situation remains stable. The change in the node with a larger radial coordinate is more obvious than the node with a smaller radial coordinate, i.e., the node with a big degree does not incur a big change in the evolution, which is consistent with our basic assumption.

### 4.5. Important Nodes Searching for Downstream Routing

#### 4.5.1. Settings

We performed performance evaluations within the downstream routing task on two datasets. One is the synthetic dataset nPSO-1, and the other is the realistic dataset DBLP. The other settings are the same as the previous evaluation.

#### 4.5.2. Metrics

*Traffic Load Centrality (TLC):* First, we use TLC to calculate the importance of nodes for routing. It assumes each node sends a unit of some commodity (e.g., traffic) to any other node. The commodity is transferred from one node to a neighbor closest to the destination. If more than one such neighbor exists, the commodity would be equally divided among them. TLC is defined as the total amount of commodities passing through a node via these exchanges [53]. The more the commodity passes, the more important the node is.

*Hyperbolic Traffic Load Centrality (HTLC):* HTLC is an approximation of TLC. It assumes each node sends a unit of some commodity to each other node, and from each node (except the destination), the commodity is equally divided to its greedy neighbors. HTLC is defined as the total amount of commodity passing through a node via these exchanges. Thus, HTLC considers greedy paths over hyperbolic coordinates instead of hop-measured shortest paths [54].

We calculate HTLC using hyperbolic coordinates obtained from each scheme, respectively. Then, we compare the top-k nodes by HTLC with that by TLC that do not need hyperbolic coordinates. The more nodes in the intersection of top-k nodes by HTLC and, therefore, by TLC, the better the scheme is.

#### 4.5.3. Results

The results of important node searching are shown in Figure 9 and Figure 10. The “prediction” means that we use HTLC to predict the top-k nodes, which is calculated by TLC. The results show that our scheme does not achieve the best result, but the difference between our scheme and the other schemes is minor. This is consistent with our target, i.e., to reduce the complexity of dynamic hyperbolic embedding while maintaining good precision for downstream tasks.

## 5. Conclusions

Time complexity in dynamic hyperbolic embedding is a big challenge for temporal complex networks. In this paper, we propose low-complexity hyperbolic embedding schemes, which aim at the evolution characteristics of temporal complex networks. To tackle the embedding efficiency problems within different dynamic network evolutionary processes; we propose two methods for medium-scale and large-scale networks. At first, we utilize the matrix perturbation approach to solve eigenvalue and eigenvector increments with low complexity in medium-scale networks. Secondly, we construct the geometric initialization to realize the low-complexity update for large-scale networks. The performance evaluations are implemented in different application scenarios. The evaluations show our schemes can model the smooth increments of a dynamic network very well. Our future work includes a robust model for dealing with dramatic increments caused by noise and outliers.

## Figures and Tables

**Figure 1 sensors-22-09306-f001:**
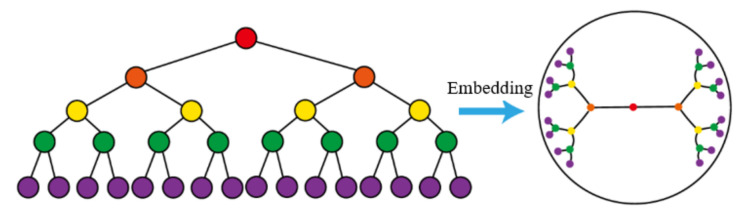
An illustration of tree structure embedding into the hyperbolic space.

**Figure 2 sensors-22-09306-f002:**
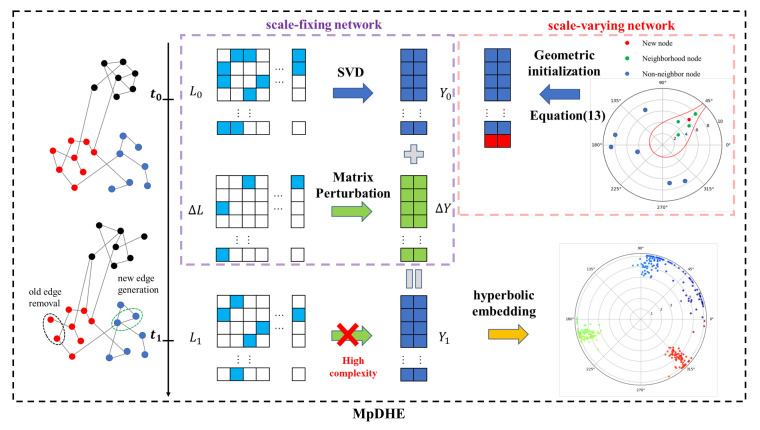
The overview of MpDHE.

**Figure 3 sensors-22-09306-f003:**
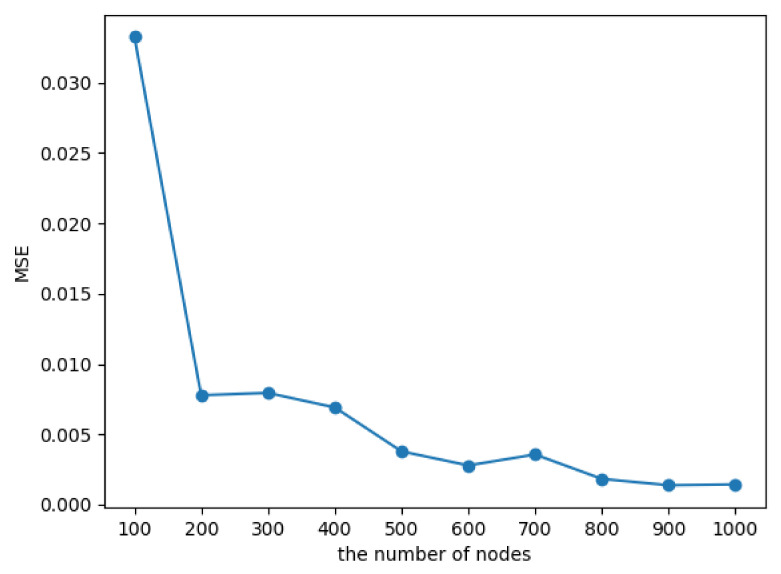
The MSE between eigenvectors of the Laplacian matrix and matrix perturbation results.

**Figure 4 sensors-22-09306-f004:**
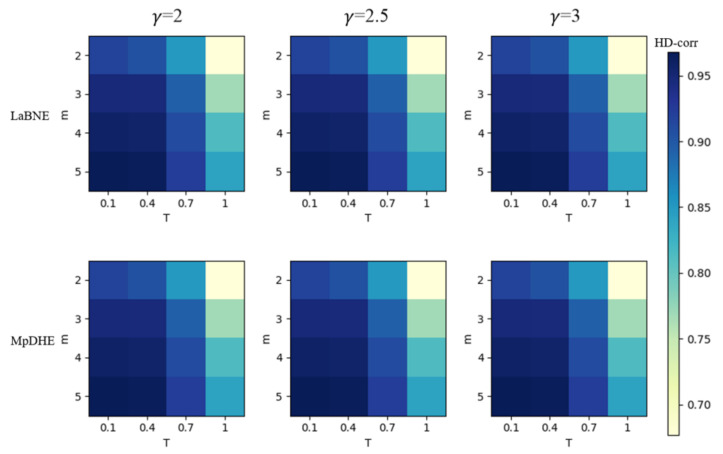
The embedding performance of networks with different parameters.

**Figure 5 sensors-22-09306-f005:**
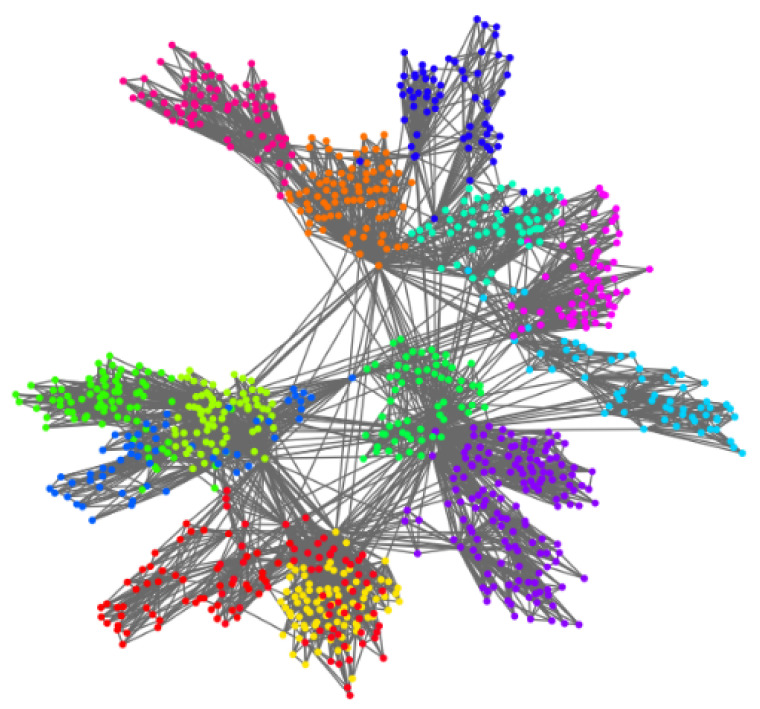
The community discovery visualization of nPSO-1 based on the Louvian algorithm.

**Figure 6 sensors-22-09306-f006:**
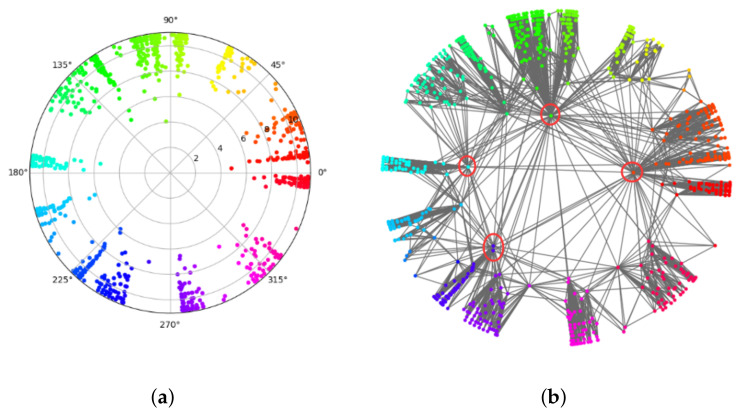
The community discovery visualization of nPSO-1 based on hyperbolic embedding. (**a**) shows the hyperbolic coordinates of the network, and (**b**) shows the result of community discovery using the CGM algorithm based on the coordinates in (**a**). We circled some key nodes which connect with many nodes, and we can see they usually have small radial coordinates in hyperbolic space.

**Figure 7 sensors-22-09306-f007:**
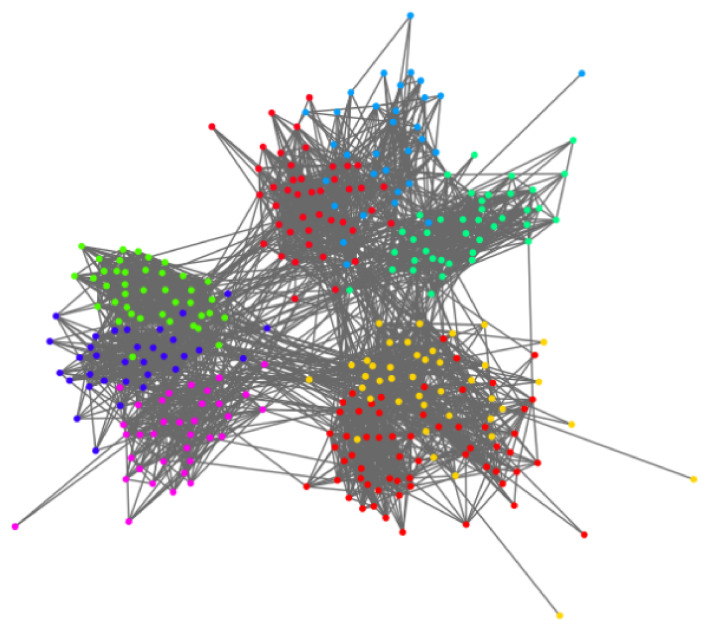
The community discovery visualization of the Students dataset based on the Louvian algorithm.

**Figure 8 sensors-22-09306-f008:**
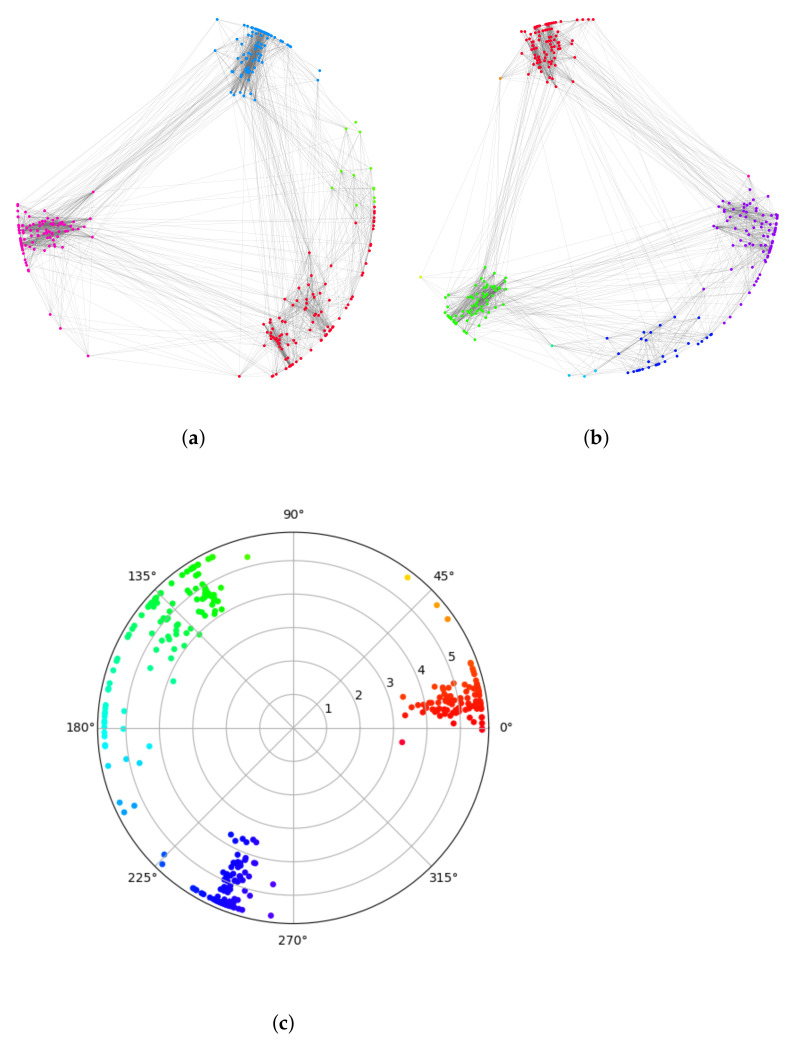
The community discovery visualization of the Students dataset based on hyperbolic embedding. (**a**) is the visualization of net0 in “Students” and (**b**) is the visualization of net1, we can see the evolution from net0 to net1. (**c**) shows the hyperbolic coordinates of the network.

**Figure 9 sensors-22-09306-f009:**
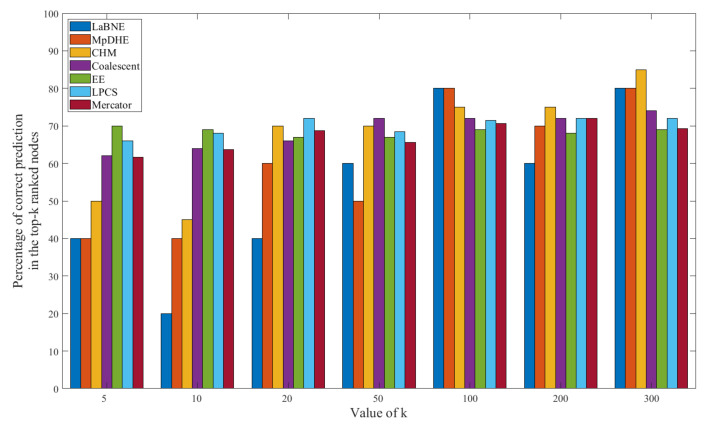
The proportion of correct predictions in top-k nodes on the nPSO-1 dataset.

**Figure 10 sensors-22-09306-f010:**
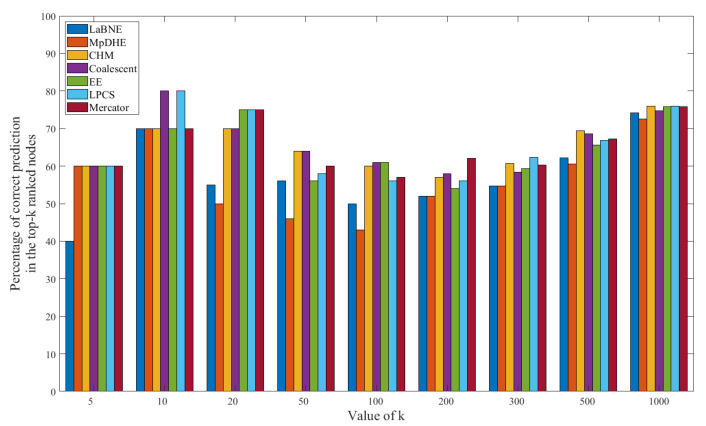
The proportion of correct predictions in top-k nodes on the DBLP dataset.

**Table 1 sensors-22-09306-t001:** The hyperbolic embedding performance for different proportions of changed nodes.

Metrics	Schemes	1%	5%	10%	15%	20%	25%	30%
HD-corr	EE	0.9368	0.9435	0.9450	0.9496	0.9466	0.9491	0.9338
	Coalescent	0.9755	0.9759	0.9754	0.9759	0.9756	0.9753	0.9759
	LPCS	0.9548	0.9547	0.9536	0.9549	0.9548	0.9544	0.9544
	CHM	0.9425	0.9429	0.9433	0.9422	0.9429	0.9430	0.9421
	Mercator	0.9432	0.9435	0.9425	0.9425	0.9428	0.9417	0.9422
	LaBNE	0.9681	0.9687	0.9684	0.9665	0.9681	0.9690	0.9680
	MpDHE	0.9679	0.9683	0.9676	0.9674	0.9670	0.9684	0.9675
C-score	EE	0.8875	0.9035	0.9369	0.9334	0.9327	0.9412	0.9217
	Coalescent	0.9942	0.9964	0.9964	0.9963	0.9962	0.9965	0.9958
	LPCS	0.9762	0.9766	0.9756	0.9763	0.9765	0.9768	0.9763
	CHM	0.9710	0.9720	0.9729	0.9714	0.9720	0.9729	0.9715
	Mercator	0.9836	0.9891	0.9856	0.9893	0.9874	0.9874	0.9879
	LaBNE	0.9920	0.9927	0.9925	0.9893	0.9936	0.9922	0.9917
	MpDHE	0.9923	0.9927	0.9918	0.9906	0.9926	0.9919	0.9918

**Table 2 sensors-22-09306-t002:** The hyperbolic embedding performance on different network scales.

Metrics	Schemes	1000	5000	10,000	12,500	15,000
HD-corr	EE	0.896560	0.944691	0.948185	0.956089	0.947904
	Coalescent	0.907575	0.974702	0.975469	0.975531	0.975494
	LPCS	0.851357	0.945676	0.955054	0.955483	0.958091
	CHM	0.857466	0.936428	0.942225	0.944796	0.946437
	Mercator	0.891975	0.947479	0.943099	0.939006	0.935306
	LaBNE	0.899086	0.967056	0.967934	0.967032	0.968298
	MpDHE	0.899059	0.967048	0.967934	0.967028	0.968298
C-score	EE	0.908284	0.923507	0.933810	0.941811	0.937041
	Coalescent	0.936283	0.992217	0.995842	0.996492	0.996578
	LPCS	0.885428	0.965513	0.977327	0.977433	0.979936
	CHM	0.888015	0.961748	0.971367	0.974466	0.976481
	Mercator	0.931286	0.989662	0.986334	0.985775	0.987029
	LaBNE	0.932585	0.989196	0.992568	0.991948	0.993236
	MpDHE	0.932524	0.989215	0.992576	0.991945	0.993233

**Table 3 sensors-22-09306-t003:** The hyperbolic embedding performance with different time steps.

Metrics	Schemes	net1	net2	net3	net4	net5
HD-corr	EE	0.949628	0.947528	0.950742	0.945694	0.947809
	Coalescent	0.975497	0.975550	0.975630	0.975471	0.975483
	LPCS	0.953126	0.955325	0.953265	0.953511	0.956482
	CHM	0.942017	0.942881	0.942082	0.942308	0.942248
	Mercator	0.942902	0.943105	0.942746	0.943036	0.942983
	LaBNE	0.966788	0.966792	0.966794	0.966767	0.966768
	MpDHE	0.966784	0.966784	0.966781	0.966752	0.966753
C-score	EE	0.940130	0.935073	0.941542	0.919566	0.936673
	Coalescent	0.996244	0.996243	0.996215	0.996252	0.996244
	LPCS	0.974311	0.977519	0.974655	0.975197	0.978727
	CHM	0.971405	0.972027	0.970815	0.971477	0.971230
	Mercator	0.987539	0.987259	0.987542	0.987485	0.987348
	LaBNE	0.991716	0.991711	0.991673	0.991694	0.991691
	MpDHE	0.991708	0.991699	0.991668	0.991693	0.991680

**Table 4 sensors-22-09306-t004:** The statistical metrics of the datasets.

Network	$\left|V(G_{0})\right|$	$\left|E(G_{0})\right|$	$\left|\gamma (G_{0})\right|$	$|E_{\mathrm{add}}|$	$|E_{\mathrm{del}}|$
nPSO-1	1000	4985	2.67	1000	0
Students	323	2942	5.67	1236	1474
BS	8796	174,836	2.99	102,243	100,882
DBLP	4442	24,734	3.75	18,369	6872
ArXiv-HepPh	5934	275,668	4.54	106,180	112,524

**Table 5 sensors-22-09306-t005:** The time for network embedding.

Dataset	nPSO-1	Students	BS	DBLP	Arxiv-Hepph
EE	2.0472	1.2613	149.1714	16.4267	158.1508
Coalescent	6.4592	8.1858	584.7455	14.6385	314.1906
LPCS	26.3493	11.0025	165.8086	98.8830	95.1393
CHM	79.0703	0.0596	1611.3348	509.7389	666.9041
Mercator	36.1439	5.6540	2001.5367	463.9197	698.7599
LaBNE	0.1789	0.0346	7.7479	2.2787	3.6021
MpDHE	**0.0515**	**0.0161**	**2.5832**	**0.6754**	**1.0156**

**Table 6 sensors-22-09306-t006:** Community discovery on dynamic networks.

Metrics	Schemes	nPSO-1	Students	BS	DBLP	Arxiv-Hepph
Modularity	EE	0.833421	0.666448	0.727889	0.250586	0.475920
	Coalescent	0.379924	0.497497	0.244465	0.177808	0.276130
	LPCS	0.684598	0.457470	0.401710	0.424134	0.439378
	CHM	0.762171	0.672836	0.758546	0.606665	0.641630
	Mercator	0.820491	0.631689	0.708975	0.538743	0.517403
	LaBNE	0.839045	0.610584	0.702995	0.465782	0.438718
	MpDHE	0.825334	0.593814	0.704195	0.438426	0.439234
Community number	EE	17	13	50	81	62
	Coalescent	146	9	533	828	442
	LPCS	9	9	17	6	9
	CHM	17	8	19	16	20
	Mercator	16	7	18	19	11
	LaBNE	25	10	57	34	20
	MpDHE	18	3	42	49	68

## Data Availability

Not applicable.

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
