# Peer review of "Low-Complexity Hyperbolic Embedding Schemes for Temporal Complex Networks"

_sensors, 2022, doi:10.3390/s22239306_

Round 1

Reviewer 1 Report

The authors present a network embedding approach in hyperbolic space coordinates for dynamic network topologies. They first apply an existing static network embedding technique called LaBNE for the embedding the network instance at the initial time.  Then they propose MpDHE that (i) updates the hyperbolic nodes coordinates using matrix perturbation on the Euclidean coordinates that is a technique that was first applied in the Euclidean space, (ii) assigns coordinates to new nodes.

The approach of the authors is interesting.

However, the simulation results need significant editing for improving the English language. The remaining text is better in this sense.

Also, about the tile Temporal complex network and initial static embedding of section 2.3, I think the part temporal complex network is not needed as it is explained in an earlier section.

Please write the algorithm at the end of page 6 as a nice-looking algorithm in a table.

Regarding the approach I have the following comments:

1. In table 1 you compare it with other hyperbolic embedding approaches in the Euclidean space. How are these approaches working? What is your novelty? Since you can directly compare your approach with them I believe they do the same thing as yours. Also, all approaches seems to have same metrics values. Why is yours significantly better?

2. Moreover, you show the application of visualization to evaluate how good your embedding is. It is known that the hyperbolic space can show and accommodate much more data than the Euclidean one as all the infinite hyperbolic space is shown within the Poincare disc in the two dimensions.  What about presenting another application such as routing? Please check also this paper that is very relevant:

E. Stai, K. Sotiropoulos, V. Karyotis and S. Papavassiliou, "Hyperbolic Embedding for Efficient Computation of Path Centralities and Adaptive Routing in Large-Scale Complex Commodity Networks," in IEEE Transactions on Network Science and Engineering, vol. 4, no. 3, pp. 140-153, 1 July-Sept. 2017, doi: 10.1109/TNSE.2017.2690258.

3. What would be the complexity difference if you were recomputing the embedding at every time instance that the network changes, i.e., using LaBNE iteratively instead of the proposed MpDHE.

Author Response

Please see the attachment。

Reviewer 2 Report

In this contribution, authors propose a modification of a method to embed networks in hyperbolic spaces, specifically the LaBNE method, that allows to analyse temporal networks. In short, it allows to modify the embedding obtained at one step taking into account the modifications in the network; the embedding is thus updated at each step, as opposed to recalculated from scratch, thus saving in computational cost. Results presented are interesting, and the benefits of the method seem clear. There are a couple of issues that should be tackled, before the paper can be published.

Major one. Taking into account that the major advantage of this method is the reduced computational cost, I found surprising that this issue is not studied. I think it would be important to compare the cost of embedding networks with different characteristics, both with the proposed approach and the LaBNE one. While the advantage is easy to see, having it in numbers would improve the value of the paper.

Second issue. While the methods are well explained, I found odd that the definition of the LaBNE method does not coincide with the original paper [24]. Specifically, see Eq. (5), which is quite different from the original one. It has a threshold, through R, which is not included in the original. Also, I could not understand what T corresponds to. Authors should clarify this point, to make it easier for readers to refer to the original paper.

The paper if OK, in terms of text, but there are many things that could be improved. Just to point a few. In lines 67/68, the word "network" is repeated 4 times, making the sentence very hard to read. In line 70, the sentence seems incomplete, especially the part "the complexity is pretty high". Another example: in Sec. 4, "European coordinates" is mentioned... I guess it is Euclidean? In general, I think the text could be revisited and improved.

Two minor issues.

Firstly, what does the authors mean by "scale-fixing complex networks"? I've never seen this expression before.

Secondly, in lines 142/143, it is said that complex networks have a tree-like structure. This is not an universal property: many networks may have, but not all. I think it would be better to rephrase this.

Round 2

Reviewer 1 Report

The authors have addressed my comments. 

Author Response

Thank you for your comments. According to your suggestion, we made thorough revisions of the paper to improve the language expression. All the revisions are highlighted in blue color in the paper. Your comments were very helpful and we did our best to follow your valuable comments and suggestions. Thank you!